

# Track cycling sprint sex differences using power data

Hamish Ferguson[1], Chris Harnish[2], Sebastian Klich[3], Kamil Michalik[4], Anna Katharina Dunst[5], Tony Zhou[1] and J Geoffrey Chase[1]

[1] Centre for Bioengineering, Department of Mechanical Engineering, University of Canterbury, Christchurch, New Zealand
[2] Department of Exercise Science, Murphy Deming College of Health Sciences, Mary Baldwin College, Fisherville, Virginia, United States
[3] Department of Paralympic Sport, Faculty of Physical Education and Sports, Wrocław University of Health and Sport Sciences, Wroclaw, Poland
[4] Department of Human Motor Skills, Faculty of Physical Education and Sports, Wrocław University of Health and Sport Sciences, Wroclaw, Poland
[5] Institute of Applied Training Science, University of Leipzig, Leipzig, Germany

Corresponding author
Hamish Ferguson,
hamish.ferguson@pg.canterbury.ac.nz

## ABSTRACT

**Objectives:** Currently, there are no data on sex differences in the power profiles in sprint track cycling. This cross-section study analyses retrospective data of female and male track sprint cyclists for sex differences. We hypothesized that women would exhibit lower peak power to weight than men, as well as demonstrate a different distribution of power durations related to sprint cycling performance.

**Design:** We used training, testing, and racing data from a publicly available online depository (www.strava.com), for 29 track sprint cyclists (eight women providing 18 datasets, and 21 men providing 54 datasets) to create sex-specific profiles. $R^2$ was used to describe model quality, and regression indices are used to compare watts per kilogram (W/kg) for each duration for both sexes against a 1:1 relationship expected for 15-s:15-s W/kg.

**Results:** We confirmed our sample were sprint cyclists, displaying higher peak and competition power than track endurance cyclists. All power profiles showed a high model quality ($R^2 \geq 0.77$). Regression indices for both sexes were similar for all durations, suggesting similar peak power and similar relationship between peak power and endurance level for both men and women (rejecting our hypothesis). The value of $R^2$ for the female sprinters showed greater variation suggesting greater differences within female sprint cyclists.

**Conclusion:** The main finding shows female sprint cyclists in this study have very similar relationships between peak power and endurance power as men. Higher variation in W/kg for women in this study than men, within these strong relationships, indicates women in this study, had greater inter-athlete variability, and may thus require more personalised training. Future work needs to be performed with larger samples, and at different levels to optimize these recommendations.

## ARTICLE SUMMARY

1) Uses a novel method of collecting open and freely available data to open up collection of athlete and event specific data.

2) Compares female data previously not studied separately for sprint cycling athletes.

3) Discusses similarities and differences by sex in track sprint cyclists, and gives guidelines how to train/coach both.

4) Limited by the use of national level cyclists identified from competition performances.

5) Future work should compare high performance cyclists by sex and include relevant supporting blood assays.

## INTRODUCTION

Track cycling is a sub-section of cycle sport, including the sprint events: individual sprint, Keirin, time trial (female 500-m and male 1,000-m), and the team sprint (*Stadnyk et al., 2021*). With the exception of the time trial, the races are similar for both sexes, and men and women compete in the same events at the Olympic Games (*Ferguson, Harnish & Chase, 2021*). Thus, it is common practice in sprint performance training, for male and female riders to train the same (*Craig & Norton, 2001*; *Wiseman, 2015*).

The authors found no studies on the sex-specific training of sprint cyclists. Although the demands of competition are similar, there are sex differences in the anatomy and physiology of elite-level track sprinters (*de Poli et al., 2019*; *Gardner et al., 2005*; *Mangine et al., 2014*; *Perez-Gomez et al., 2008*; *Yanagiya et al., 2003*). Differences between sexes were observed in muscle fiber distribution, fiber size, and levels of succinate dehydrogenase, lactic dehydrogenase, and phosphorylase of sprint runners (*Costill et al., 1976*). Similarly, sex differences were observed in running 100 and 200-m sprinters measuring accumulated oxygen deficiency, but not lactate/phosphocreatine measures (*Duffield, Dawson & Goodman, 2004*). Female athletes showed less fatigue than males in knee extensor contraction, which was independent of strength (*Ansdell et al., 2017*). Females had a higher critical intensity relative to maximal force than males, and this difference was observed above and below a metabolic threshold (*Ansdell et al., 2019*). While there were no differences in fatigue along the power-duration curve, relative to a maximal ramp test, the mechanisms of fatigue differed between sex (*Ansdell et al., 2020a*). Aerobic contribution to a 30-s Wingate Anaerobic Test (WAnT) was 20% for men, but increased to 25% for women (*Hill & Smith, 1993*).

With the challenges of obtaining physiological measures in the laboratory, let alone in the field, the use of various models based on performance data from either velocity or power data have been proposed to explain sprinting performance (*Dunst & Grüeneberger, 2021*; *Dunst, Grüeneberger & Holmberg, 2021*; *Ferguson et al., 2021*). However, these models do not always capture the performance outcome adequately (*Zaccagni et al., 2019*). Hence, there remains a need to consider data and models accounting for sex specific response's to training, and quantifying these differences to better guide training approaches.

A recent study has tracked the progression of six male cyclists over a season (*Desgorces et al., 2023*), however there is no research modelling performance of female sprint cyclists over a season. Therefore, this study retrospectively analyses open-source, publicly available power meter data of elite male and female track cyclists to identify sex differences in power, performance, and potentially energy system involvement (ATP-phosphocreatine, glycolytic and oxidative; *Martin, Davidson & Pardyjak, 2007*). Based on observed physiological differences, we hypothesise women will display differences in the short duration power curve, which could have meaningful implications on training for female sprint cyclists.

## METHODS

### Participants

We used the open source Strava website (www.strava.com) to identify sprint cyclists competing in New Zealand, who posted both their racing, testing and training data. We harvested Strava data from 29 track sprint cyclists (eight women and 21 men). The use of a single, open source site ensures all data was stored similarly, and any computations used similar data structures and density. Data was downloaded from the Strava app according to the Strava Privacy Policy (https://www.strava.com/legal/privacy).
No personal information was taken from the Strava site. The Sauce extension (https://www.sauce.llc/) was used to download a .tcx format file containing power meter data for each set of rider files. We downloaded ride files from competition and training sessions for 3–12 months prior and incorporating a NZ Championship, or World Master's Championship event. The University of Canterbury, Christchurch, New Zealand Human Research Ethics Committee gave an exemption for ethics approval due to the publicly available nature of the data (2022/06/EX).

Table 1 describes the participants for this study, and also the endurance data from *Ferguson et al. (2021)*. The data from *Ferguson et al. (2021)* was solely used to assure our sample were sprint cyclists by comparing them to track cycling endurance cyclists. Retrospective power meter data during training and racing from a group of eight women provided 18 datasets, and 21 men provided 54 datasets. The inclusion criteria for classification as a sprint cyclist was the ability to place top four in the match sprint, Keirin, and track time trial events at New Zealand championship events, competed at the Junior World level in sprint competition, and top 4 placing in Masters World Championship events.

Our final inclusion criteria were data reflecting a maximal effort for shorter than competition durations (1–5 s), competition durations (15–30 s) and longer durations (45-s–2-min). Specifically: 1, 5, 10, 15, 30, 45, 60-s, and 2-min, over a minimum of 3 months, and a maximum of 12 months. Data included maximal efforts over all durations from racing, testing, and training sessions.

### Study overview

This is an exploratory study using retrospective data to present data from female sprint cyclists. With data collected from periods including testing, training and racing sessions,

**Table 1 Participant data included this study with sprint data sets in the first set and the validation endurance (END) cyclist data sets and demographics in the second.**

| | All: median (IQR) | Female: median (IQR) | Male: median (IQR) |
|---|---|---|---|
| Sprint data sets | | | |
| Data sets | 72 | 18 | 54 |
| Participants | 29 | 8 | 21 |
| Age (years) | 30 [17–41] | 23 [17–27] | 33 [17–43] |
| Weight (kg) | 77.9 [75.0–84.5] | 65.9 [60.0–73.8] | 82.0 [75.0–88.0] |
| Endurance (END) data sets | | | |
| END data sets | 144 | 16 | 128 |
| Participants | 69 | 9 | 60 |
| END (Age) | 26 [17–38] | 19 [17–30] | 27 [17–39] |
| END (Weight) | 70.0 [65.0–75.5] | 57.0 [56.0–63.5] | 72.0 [67.0–76.0] |

Notes:
The median and interquartile range (IQR) are presented.
Participant data for the female and male sprint and endurance cyclists.

focused on sprint cycling competition performance. Thus, peak power was collected for 1, 5, 10, 15, 30, 45, 60-s, and 2-min durations. These data were analysed to determine relationships between all durations and 15-s peak power, which reflects the power demands of the flying 200-m, match sprint and the first rider in the team sprint (*Ferguson, Harnish & Chase, 2021*). While some sprints may take longer than 15-s, up to 30–45-s depending on events and tactics, the 15-s duration best reflects the nature of sprint cycling and is the comparator in this study.

## Analysis

Peak power values for all durations are compared to those at 15-s to generate a power curve between 15-s and the input duration. Model quality is assessed using the total least squares correlation coefficient, $R^2$, which accounts for variability and error in both variables (*Golub & Van Loan, 1980*; *Markovsky & Van Huffel, 2007*). Higher $R^2$ values indicate stronger relationships, and thus, a better model and predictor.

Linear model slope indicates the trade-off between power for a given duration and 15-s. Each slope is assessed against the 1:1 relationship found where 15-s and 15-s are compared, where the difference from 1:1 shows the trade-off between two durations. Our model includes the point (0,0), a point of null power. This assumption is physiologically relevant, where a rider could not have a measurable W/kg of zero.

The two-tailed unpaired t-test with Welch's correction was used to compare the differences in power between the sprint cyclist data and endurance groups, and thus to ensure our analysis group was truly comprised of sprint cyclists. A similar test was performed to measure differences between the male and female groups at each duration using 15-s as a dependent variable. A difference was considered to be significant as $P < 0.05$. We used MatLab version R2022a (The MathWorks, Natick, MA, USA) to analyse the data. Our hypothesis states we expect a difference between male and female cyclists in

**Table 2 Independent samples T-Test for sprint and endurance cyclists for 1, 15, 30, 45, 60-s and 2-min.** The table shows *P*-values for each duration highlighting how the sprint athlete group are more powerful than endurance athletes.

|  | Statistic | df | P |
|---|---|---|---|
| W/kg 1-s | −5.49 | 10.4 | <0.001 |
| W/kg 15-s | −4.19 | 10.2 | 0.002 |
| W/kg 30-s | −3.31 | 10.6 | 0.007 |
| W/kg 45-s | −2.52 | 11.2 | 0.028 |
| W/kg 60-s | −3.14 | 11.2 | 0.009 |
| W/kg 2-min | −3.21 | 12.2 | 0.007 |

this latter set of comparisons, and specifically different slopes for these models, reflecting differences in physiology.

# RESULTS

## Assuring a sprint cohort

Before we carried out analysis of our data, we confirmed our sample were track sprint cyclists by comparing their data against a sample of track cycling endurance cyclists (pursuit and mass start bunch events) data collected in *Ferguson et al. (2021)*. Table 2 displays the t-test for sprint *vs* endurance cyclists to show our sample of sprint cyclists have higher short-term sprint power and top-end aerobic power than track and road endurance cyclists. As expected, as the power duration extends to 2-min the endurance cyclists display higher maximum mean power, further validating this selection of sprint cyclists against endurance cyclists. $P \leq 0.05$ for all durations show our sprint group are different from the endurance rider group.

## Comparing male and female sprint cyclists

Table 3 summarises the power (W/kg) data for all subjects. It shows the power for each duration, and includes power for 1, 15, 30-s and 2-min durations for endurance cyclists (*Ferguson et al., 2021*). Over shorter durations sprint cyclists have a higher W/kg than endurance cyclists, as expected between these two groups. For the 2-min duration we see the endurance cyclists have a higher W/kg, again, as expected. For all durations, both sprint and endurance cyclists, female athletes have a lower W/kg than male athletes. As noted, all these outcomes match expectations.

Table 4 lists the $R^2$ value and slope comparing 15-s power (W/kg) against each of the durations. $R^2$ values were different between female and male athletes, while the slopes between sexes were similar. Figure 1 illustrates both slope and $R^2$ values for each of the durations in this study. This figure visually describes the data from Table 4 and illustrates the different spread of data for female sprinters compared to males, and the similarity of the slopes between sprinters of each sex.

Finally, Fig. 2 plots the differences for all durations between female and male sprint cyclists, and also between female and male endurance athletes for selected durations. This

**Table 3 Descriptive data for both female and male, female, and male athletes for all durations, and in red the endurance athletes for 1, 15, 30-s, and 2-min.** Median values and the interquartile range.

| | BOTH (Median and IQR) | FEMALE (Median and IQR) | MALE (Median and IQR) |
|---|---|---|---|
| 1-s W/kg | 18.41 [19.69–16.82] | 16.97 [17.57–16.01] | 19.00 [20.43–17.83] |
| END 1-s W/kg | 15.53 [19.08–15.67] | 15.41 [15.63–14.55] | 17.78 [19.25–16.31] |
| 5-s W/kg | 17.03 [18.69–15.51] | 15.48 [16.26–14.15] | 17.66 [19.29–16.35] |
| 10-s W/kg | 15.44 [17.20–13.68] | 13.86 [14.56–12.57] | 16.08 [17.34–14.60] |
| 15-s W/kg | 14.38 [16.32–13.29] | 13.28 [14.19–12.21] | 15.46 [16.71–14.15] |
| END 15-s W/kg | 13.27 [14.59–11.88] | 11.65 [12.26–10.29] | 13.44 [14.71–12.11] |
| 30-s W/kg | 10.75 [11.55–9.69] | 9.80 [10.70–9.12] | 11.13 [11.76–10.00] |
| END 30-s W/kg | 10.61 [9.86–12.03] | 9.54 [10.28–9.09] | 10.91 [12.05–9.93] |
| 45-s W/kg | 8.68 [9.45–7.73] | 7.89 [8.17–8.17] | 9.00 [9.58–8.22] |
| 60-s W/kg | 7.51 [8.04–6.68] | 6.85 [7.31–5.94] | 7.78 [8.08–7.14] |
| 2-min W/kg | 5.57 [6.10–4.74] | 5.42 [5.88–4.74] | 5.63 [6.39–4.82] |
| END 2-min W/kg | 6.64 [7.20–6.15] | 6.20 [6.58–5.91] | 6.69 [7.30–6.21] |

**Table 4 $R^2$ and slopes for 15 and 30 s power and power at all durations.** The table highlights the differences in relationships between female and male sprint cyclists. The slopes show there is a little difference between female and male sprint cyclists.

| Time | Female | | Male | |
|---|---|---|---|---|
| | $R^2$ 15-s | Slope 15-s | $R^2$ 15-s | Slope 15-s |
| 1-s | 0.85 | 0.80 | 0.92 | 0.84 |
| 5-s | 0.91 | 0.84 | 0.96 | 0.89 |
| 10-s | 0.98 | 0.94 | 0.99 | 0.94 |
| 15-s | 1.00 | 1.00 | 1.00 | 1.00 |
| 30-s | 0.79 | 1.34 | 0.92 | 1.39 |
| 45-s | 0.77 | 1.68 | 0.89 | 1.68 |
| 60-s | 0.77 | 1.93 | 0.90 | 2.03 |
| 2-min | 0.78 | 2.53 | 0.88 | 2.67 |

illustrates the similarities between female and male W/kg sprinters reflected in the slopes in Table 4, and shows how sprint cyclists have better short-term power at 1–10-s, are comparable at 30-s, and at 2-min endurance cyclists have a higher W/kg.

# DISCUSSION

The purpose of this data was to compare power output between female and male track sprint cyclists to assess sex-based differences. Field-based power data from a group of well-trained male and female sprint cyclists was retrospectively analysed, to identify sex differences in power. We found no differences between the slopes of the data through all durations measured against 15-s power; however, we did find differences in $R^2$ between female and male sprint cyclists suggesting greater variability within female sprint cyclists.

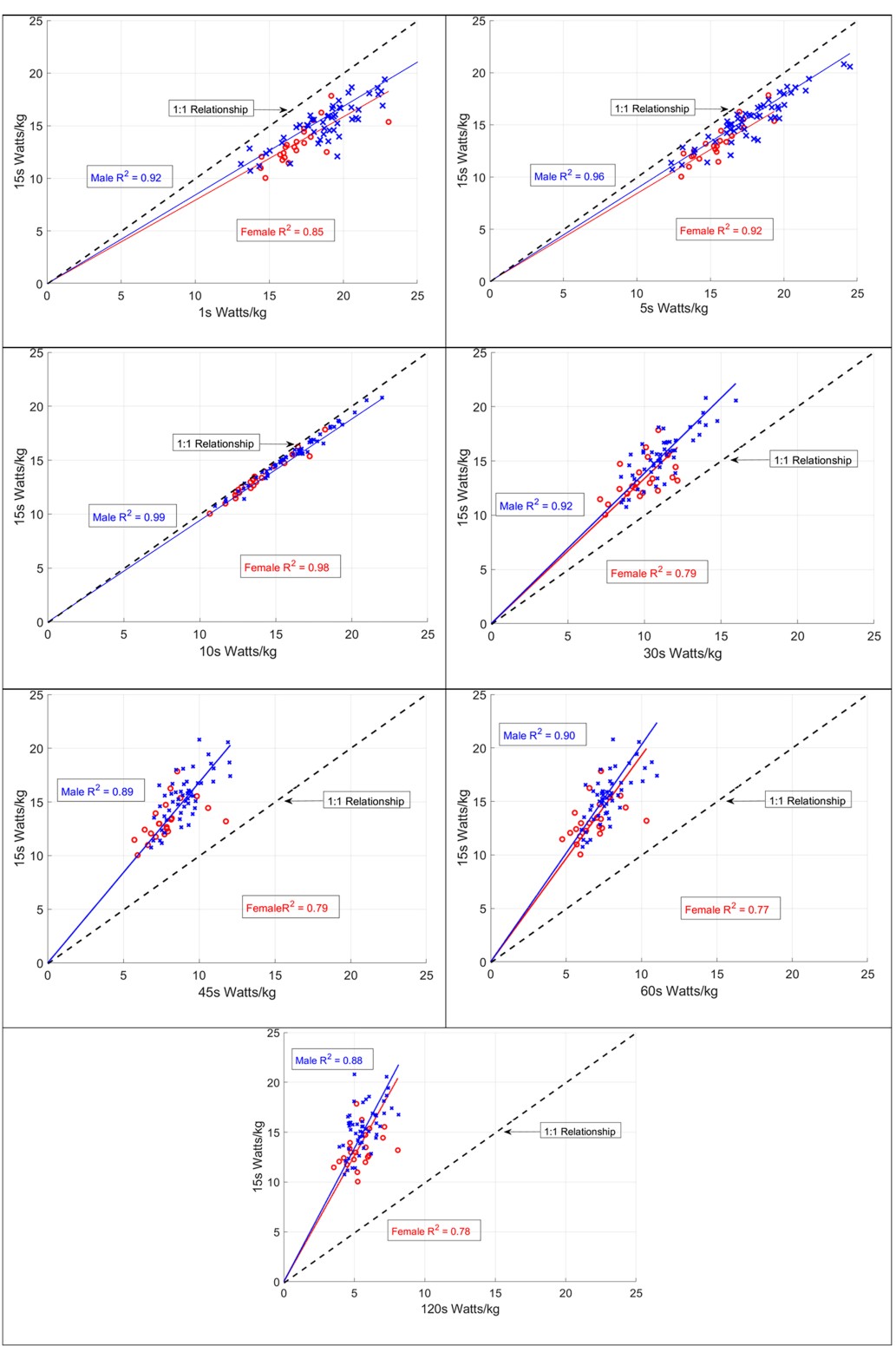

**Figure 1** **Slope and R² for 15-s W/kg and all durations in this study.** We describe the slope and R² for all durations against 15-s watts per kilogrammes. This describes the close relationships between male and female sprint athletes, and the variation observed in female sprint athletes.

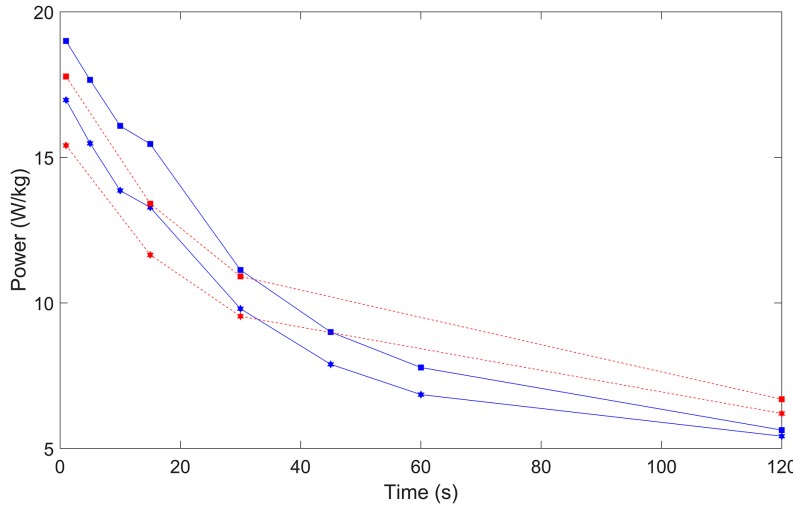

**Figure 2 The comparison of power (W/kg) for female (hexagon) and male (square) sprint (blue line) and endurance (red dotted line) track cyclists from 1-s to 2-min measuring peak power.** This figure describes the differences in power between male and female sprint cyclists highlighting greater differences in short term durations. It also compares endurance athlete power with sprint athlete power. This highlights how endurance athletes short term power is lower than sprint athletes; however, as duration increases, the endurance athletes display more power.

The slope of the lines suggests the relationship between each duration and 15-s W/kg is similar for males and females with no notable differences. Thus, both female and male sprint cyclists should equally focus on both peak power, power specific to competition durations, and power over competition duration as each has a positive relationship with competition specific power durations of 15–30 s. As a second main result, while the $R^2$ values for the data show a strong relationship for both male and female data (Table 4 and $R^2 \geq 0.77$ in all cases), the female data is more variable than the male data based on lower $R^2$ values for all durations except 30-s. This result highlights greater intra-individual variability among female athletes, despite similar power curves, and thus the potential corresponding need to individualise training for women as they vary more in how they relate to the line of best fit. Female athletes above the line will benefit from training their capacity to hold event-specific power for longer, while female athletes below the line of best fit would benefit from training to improve their event specific power.

A summary of the data for each duration for the sprinters, and selected data for the endurance riders, provides confirmation of the differences between sprint and endurance cyclists. The explanation for differences in respiratory system, circulatory system, supraspinal neurons, motoneurons and skeletal muscle (*Ansdell et al., 2020b*; *Miller et al., 2018*; *Nuell et al., 2019*). A second common distinction is anatomical differences which take shape as females mature from adolescence to adulthood (*Doré et al., 2001*; *Doré, Bedu & Van Praagh, 2008*). The data showed for 1, 5, 10, 15, 30, and 60-s the W/kg for the sprint cycling group was higher than the endurance group, consistent with previous research (*Kordi et al., 2020*). For 2-min, the endurance group had a higher W/kg. This is expected, due to the higher aerobic contribution to performance in endurance athletes (*Billat et al.,*
*2009*; *Withers et al., 1991*). Thus, overall, this analysis ensured the main analysis comprised only sprint cyclists.

The limitations of this research is elite athletes will not only focus on sprint training and sprint cycle racing. It is expected riders in this study would perform track endurance events and even short and flat road races. The wide range of age groups does also lead to variability in the data, however crossover between female and male sprint athletes is not too distinct. Elite sprint cyclists in a nationally funded programme would be expected to be focused more on sprint-specific training and racing. In a case study of New Zealand elite cyclists preparing for the 2012 Olympic Games their training was purely focused on sprint performance (*Wiseman, 2015*). However, testing metrics used in the preparation suggesting performance was progressing towards the Olympics was not reflected in actual performance in competition (*Ferguson, Harnish & Chase, 2021*).

We used data harvested from an open online source (Strava, www.strava.com). Athletes can elect to upload their data to Strava privately and can remove the power data component from viewing. It is also notable that we found no data, public and private for high performance cyclists (NZ and International). Public posting of this data could actually assist in efforts to monitor high performance cycling to control the use of illegal drug taking in sport (*Puchowicz et al., 2018*; *Schumacher & Pottgiesser, 2009*). There is an imbalance between women and men doing sprint cycling reflected in the numbers sprinting, and also in the sprinter *vs* endurance rider where more riders will do endurance events with options on both road and track.

The main application of this research is to ensure when developing the performance of female sprinters is to address the within-female individual differences, as these are wider than those seen in men. In this study, the differences within female sprint cyclists showed testing is needed to determine if females needed to train to lift their power, or to train to hold the power they could sustain for longer. Once data had been collected from a group of sprinters to get a good line of best fit. Sprinters above the line should focus on the development of capacity, and those below the line should focus on developing the ability to hold a high power for longer sprint durations. This strategy will ensure the athlete has both the power needed to perform at their best in competition, balanced with the capacity.

Future research should focus on using larger sample sizes. This will allow for better assessment of the intra-individual differences observed in this study. Data should be gathered from trained, well trained, elite (national) and high performance (international) female track cyclists to compare differences and test for the effects of different training and different racing practices (high performance tend to only do sprint events), while elite and trained will also compete in track endurance and possibly road cycling events. Our study used field based power measures to assess performance. Other performance measures such as speed and cadence could be used, and physiological measures such as lactate could shed further light on the topic.

## CONCLUSION

This is the first study to present data on female track sprint cyclists measuring peak power output from a training and racing. Data was collected for both male and female sprint

cyclists. This data was compared with data from track endurance cyclist power meter data to ensure the sample consisted of sprint athletes. Comparison of female and male sprint cyclists showed similar positive slopes suggesting all sprint cyclists would benefit from training not only peak power and competition power, but also maximal power up to 2-min based on the findings of this study. $R^2$ for all durations was lower for females suggesting greater intra-individual differences and highlight a need to be more specific when designing a programme for females to optimize their training.

### Funding
The authors received no funding for this work.

### Competing Interests
The authors declare that they have no competing interests.

### Author Contributions
- Hamish Ferguson conceived and designed the experiments, performed the experiments, analyzed the data, prepared figures and/or tables, authored or reviewed drafts of the article, and approved the final draft.
- Chris Harnish conceived and designed the experiments, analyzed the data, prepared figures and/or tables, authored or reviewed drafts of the article, and approved the final draft.
- Sebastian Klich conceived and designed the experiments, authored or reviewed drafts of the article, and approved the final draft.
- Kamil Michalik conceived and designed the experiments, authored or reviewed drafts of the article, and approved the final draft.
- Anna Katharina Dunst conceived and designed the experiments, authored or reviewed drafts of the article, and approved the final draft.
- Tony Zhou conceived and designed the experiments, analyzed the data, authored or reviewed drafts of the article, and approved the final draft.
- J. Geoffrey Chase conceived and designed the experiments, analyzed the data, prepared figures and/or tables, authored or reviewed drafts of the article, and approved the final draft.

### Data Availability
All data are available at OSF: Ferguson, Hamish. 2022. "Hamish Ferguson Sprint Cycling Data." OSF. December 29. osf.io/2y3kr.

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
