# Peer review of "Track cycling sprint sex differences using power data"

_PeerJ, doi:10.7717/peerj.15671_

## Round 0.1 · original submission · Major Revisions

The reviewers noted the importance of the study to better define women's characteristics. There's very valuable feedback in the three expert reviews provided to clarify and precise the rationale throughout.

·

Basic reporting

This is an interesting study, taking a novel approach with cycling data. Aspects could be improved throughout, which I outline below.

Experimental design

A novel approach has been taken here, by I do have an important question about the independence of data samples.

Validity of the findings

A good array of data is provided, that will be of interest to the field. Tables could be improved.

Additional comments

L15: change ‘the slope of the line of best fit as’ to ‘regression indices were‘.

L16: delete ‘me’ from ‘kilogram’.

L19: Results. This section would be enhanced by providing some data to compliment the narrative, along with supporting statistics. Furthermore, the results do show higher power outputs in male riders, confirmed also in the discussion. Similar relationships yes, but peak power was different. Please revise.

L22: change ‘more’ to ‘greater’.

L32: list keywords in alphabetical order and avoid repeating any that are included in the title.
Introduction. Writing of the second paragraph could be improved; on L58 and 60, ‘sex differences’ are mentioned but what is different between sexes, is not explained. Furthermore, there are two isolated sentences that begin on L62 and 67, both of which make a good point with relevant citations, but each could be better integrated into the narrative. Finally, L67, what was the difference between the sexes? Tell the reader that ‘females were more fatigue resistant, which is partly explained by the points above’.

L78 & 89: remove the citations within the brackets.

L92: specifically, was anything different between the male and female adolescent riders?

L96: change ‘adequately outcome’ to ‘the performance outcome adequately’.

L109: insert ‘in’ after ‘differences’.

L121: it would be worth mentioning Stava regulations here; does uploading data allow it to be shared more widely without further consent?

L126: delete the citation within the bracket.

L130: delete the additional space at the sentence end.

L156: data are pleural; subsequently, change ‘This data was’ to ‘These data were’.

L175: Statistics. Given repeated data sets are recorded from individuals, independence of samples has been violated; how has this been controlled for?

L188: Table 2 should include the W/kg data as without, the statistical comparisons are meaningless. Thus, it would be ideal to combine Table 2 and 3, or write the W/kg results into this paragraph.

L227: the discussion could be written in a more mechanistic manner, similar to the approach in the introduction. How do the current data compare to previous work and what are the mechanisms at play contributing to similarities or differences.

L257: this section would benefit from the inclusion of a recent review (Ansdell et al., 2020, Exp. Physiol) providing an overview of the sex differences underpinning potential training adaptations

L295: Also, the hormonal profile of female athletes would help to provide a mechanistic insight into training responses. Such data could, in part, explain the increased variability observed in the female data.

·

Basic reporting

See attached please

Experimental design

Fine

Validity of the findings

Good.

Additional comments

See attached.

·

Basic reporting

- The language used is clear and understandable enough.

- However, I think the link between the intersex physiological differences presented and the training method must be more precise. You suggest training sprint cyclists sould be sex-specific but what would you recommand for both sex ?

- What is te purpose of the study ? To show potential power profile differences between men and women and try to justify it by sex physiological differences or to do training recommandations ?

- About the figure 2, we can guess differences in power between male and female but it doesn’t clearly show (objectively) the greater differences in short term durations you mention. Maybe, a figure showing differences values of results between men and women would be more appropriate.

Experimental design

This study aimed to compare the peak power relationships between women and men track cyclists. Authors hypothetised a lower expression of peak power level for women, mainly in short duration efforts.

The results showed lower maximal power output at each time duration of exercise in women but a similar relationships between male and females in both sprint and endurance groups.

This is an exploratory study using retrospectively field datas, with a nice method, clearly described and an interesting purpose but not clear enough. Moreover, you mention the energy system described by Gastin (2001) to justificate your range of data choosen which can be, in my mind, questionable and taken out of the context of the field.

You suggest to individualise training more for women than men because of a higher intra-individual variability and, in my opinion, these recommandations show one of the limits of this study.

Indeed, age differences between male and female groups could influence the results. Is the power profile influenced by the experience, accepting older athletes are more experienced than younger, and reduce intra-individual variability ? Or power profile influenced by age, suspecting a decrease in short duration performance in master athletes by a nervous system deterioration with age ?

Validity of the findings

Authors used a T-test to define their sprint and endurance group, but I think a multivariate statistial analysis could be interesting to discriminate population (sex, age, etc.) Also, R² is influenced by sample size which is very different between men and women and could be cited in limits.

What is the interest of showing only power datas in W/kg and not using the absolute values ? Considering the track is flat, is the weight not so important in the performance ? Would the peak power differences and peak power relationship between men and women be different ?

I think this article would fit for the journal. The subject is in the air of time but the question and purpose need to be clearer which would help you analysis.

---

## Round 0.2 · accepted · Accept

Thank you for addressing the reviewers' comments and questions, who all pointed out the quality of your work with applicability to real-world settings. It is also a very enjoyable read.

·

Basic reporting

Well written throughout, providing an interesting dataset.

Experimental design

Well explained, sufficient to the question.

Validity of the findings

Well presented.

Additional comments

The authors have responded to my initial comments and I have nothing left to address. This study poses and interesting question and uses relevant methods to provide an interesting data set. The results will be of interest to the scientific community and the design of the experiment will be inspiring for others to adopt.

·

Basic reporting

The revisions are solid. Nice work by all.

Experimental design

Good.

Validity of the findings

Good

Additional comments

None

·

Basic reporting

Thank you to let me review this paper and for your interesting answers.
The introduction and discussion have been both clarified and well designed, taking in considerations different points of views from reviewers.

The question is clear and the results are well discussed, showing the interest for cycling coaches and scientist to use this analysis approach to assess their athletes.

Experimental design

The experimental design is fine. The research question is weel definied and the approach used (strava datas) is original.

Validity of the findings

The choice of the statistical method used was weel argued as well as than answers to my questions.

Additional comments

This article is pleasant to read and would fit with the journal.